# Evaluating Field-Collected Populations of *Cotesia flavipes* (Hymenoptera: Braconidae): Enhancing Biological Traits and Flight Activity for Improved Laboratory Mass Rearing

**DOI:** 10.3390/insects16060571

**Published:** 2025-05-28

**Authors:** Eder de Oliveira Cabral, Josy Aparecida dos Santos, Agda Braghini, Vinícius de Oliveira Lima, Enes Pereira Barbosa, Alessandra Marieli Vacari

**Affiliations:** 1Laboratory of Entomology, University of Franca (UNIFRAN), Franca 14404-600, SP, Brazil; edercabral@hotmail.com (E.d.O.C.); josysantoscosta15@gmail.com (J.A.d.S.); agdabraguine@hotmail.com (A.B.); volima_2013@hotmail.com (V.d.O.L.); enes.organicos@gmail.com (E.P.B.); 2São Martinho Mill, Pradópolis 14850-000, SP, Brazil; 3Caso do Café, Franca 14406-076, SP, Brazil; 4Emater, Claraval 37997-000, MG, Brazil

**Keywords:** biological control, integrated pest management, sugarcane crop

## Abstract

Many insect production facilities raise the parasitoid wasp *Cotesia flavipes* to help control agricultural pests. To improve the quality of these wasps, some facilities introduce new populations collected from the wild, even without knowing if they will perform better. This study compared a field-collected population of *C. flavipes* from Brazil with a population that has been reared in the laboratory for over 40 years. The goal was to determine if the field population had better biological traits and flight ability, making it a good candidate for mass production. Over 20 generations, researchers observed that the field population developed slightly faster and, by the twentieth generation, produced more offspring per parasitized host. However, both populations showed similar movement ability, with more wasps walking rather than flying. By the tenth generation, the field population had stabilized in the laboratory, and overall quality became similar to the long-term lab population. These findings suggest that introducing field-collected wasps into mass production can be beneficial, but the differences in performance diminish after several generations in captivity.

## 1. Introduction

Brazil is recognized as the leading global producer of sugar cane, accounting for more than half of the world’s sugar market. The country boasts an extensive cultivation area covering around 8.7 million hectares, resulting in a significant production of approximately 678.7 million tons of sugar cane during the 2024/2025 season [1]. This vast quantity of sugar cane production yields an impressive output of 44.0 million tons of sugar and a substantial 36.08 billion liters of ethanol [1].

There are numerous pest species that inflict economic losses on sugarcane producers. Among them, the sugarcane borer, *Diatraea saccharalis* Fabricius, 1794 (Lepidoptera: Crambidae), stands out as the most significant due to its widespread occurrence, high biotic potential, adaptation to favorable climates, protective habits against natural enemies, and, most notably, the extensive damage it causes [2]. This pest is widely distributed across sugarcane regions in Brazil and various countries in South, Central, and North America [3,4]. In addition to sugarcane, it infests several grass species, including sorghum, corn, and rice [5,6].

The direct damage inflicted by *D. saccharalis* larvae stems from their feeding habits, particularly their attack on sugarcane plants. The galleries they create within the stem cause weight loss, weaken the plant, making it susceptible to lodging and eventually leading to death or breakage [7]. Indirect losses arise from the colonization of fungi. The fungus *Fusarium verticillioides*, the agent responsible for the occurrence of the red rot complex, occurs in association with the sugarcane borer *D. saccharalis*. These pathogens induce sucrose inversion and reduce the purity of the juice, resulting in decreased harvest yields for both the agricultural and industrial sectors [8].

Considering the pest’s biology and the extensive areas dedicated to continuous sugarcane cultivation, chemical control methods prove inefficient. The larvae primarily reside within the plant during their developmental stage, making them inaccessible to conventional insecticide applications. Additionally, chemical control measures can be economically burdensome, particularly given the scale of the infested area, and may pose environmental risks [9].

In an effort to address the inefficiency of *D. saccharalis* control in Brazil, *Cotesia flavipes* Cameron, 1891 (Hymenoptera: Braconidae) from Trinidad was introduced into the state of Alagoas in April 1974, marking the commencement of this parasitoid’s involvement in the National Biological Control Program for *Diatraea* spp. in Brazil [10]. However, the parasitism results fell below expectations, particularly in the center-south region of the country [11]. Consequently, new strains of *C. flavipes* were introduced in 1978 from India and Pakistan, where the climate was similar to that of the State of São Paulo [12].

Presently, the field release of the parasitoid *C. flavipes* is conducted at a density of 6000 parasitoids per hectare. In 2007, the parasitoid was released across 800 thousand hectares of sugarcane in Brazil [13]. More recently, the parasitoid has been introduced to just over 3.5 million hectares [14]. To meet this demand, 28 private laboratories and sugar and alcohol plants have been engaged in the production of *C. flavipes*, with the parasitoid registered with the Ministry of Agriculture, Livestock, and Supply [15]. Approximately 80% of these laboratories are situated in São Paulo. Utilizing *C. flavipes* for *D. saccharalis* control in sugarcane represents one of the largest biological control programs globally, given the extensive treated area.

In the current Brazilian scenario, populations of *C. flavipes* were originally established by a small founding group, with no additional reports of wild strains being introduced. While mating occurs randomly within populations maintained in biofactories, it does not contribute to an increase in genetic diversity because all individuals are related, sharing some common ancestors [16]. Thus, the Brazilian population can be viewed as a single entity of *C. flavipes*, despite being divided into subpopulations across different biofactories. Since their introduction in Brazil, the quality of the parasitoids has not been consistently monitored. Consequently, it remains uncertain whether this species has experienced a decline in quality over time. However, studies have indicated that over ten generations, the parasitoid does not experience a decrease in quality [17]. However, other research has shown considerable differences in the size of the parasitoids, offspring production, and flight activity when comparing populations of *C. flavipes* from various Brazilian biofactories [18]. Moreover, it has been observed that the parasitoid, which previously exhibited a flight capacity of approximately 34 m, now has a maximum flight capacity of 25 m [19]. The finding that populations of *C. flavipes* studied by Botelho et al. [20] showed a dispersion of 34.38 m from the release point may suggest that, in the present era, after 50 years, *C. flavipes* may have lost the genetic traits associated with aggressiveness and dispersal potential.

In this context, with a focus on enhancing the quality of laboratory populations, a common practice in *C. flavipes* biofactories involves collecting the parasitoid from the field, rearing it in the laboratory for several generations, and then introducing it into mass rearing. However, the quality of parasitoid populations collected from the field is not typically evaluated. Therefore, the objective of this study was to determine whether a field-collected population of the parasitoid *C. flavipes* exhibits superior biological characteristics and flight activity, with the aim of integrating it into laboratory mass production to enhance the quality of parasitoids produced in biofactories.

## 2. Materials and Methods

### 2.1. Diatraea saccharalis Host Rearing

The insects were sourced from a colony established from larvae collected in a sugarcane field in April 1979 (21°21′23″ S, 48°3′48″ W) and housed in the laboratory at São Martinho mill located in Pradópolis, SP, Brazil. The larvae were reared on an artificial diet in flat-bottomed glass tubes (2.5 cm in diameter × 8.5 cm in height) until they reached the pupal stage. The composition of the artificial diet used for larvae rearing is detailed by Parra and Mihsfeldt [21]. Pupae were then removed from the diet and stored in plastic boxes (11 cm long × 11 cm wide × 3.5 cm high) lined with filter paper until adult emergence. The moths were subsequently transferred and housed in cylindrical PVC cages (20 cm in diameter × 20 cm in height), covered at the top with voile fabric, and internally lined with white bond paper to serve as a substrate for oviposition. Each cage accommodated 40 females and 45 males, and no additional diet was provided for the adults. A water container of suitable size was placed at the bottom of each cage to maintain humidity, and a screen prevented the insects from escaping. Daily, the paper containing eggs was replaced, and new paper was added to the adult cage. Prior to insertion, the egg-laden paper was treated with a copper sulfate solution (1% *v*/*v*), dried at room temperature, and then the egg masses were cut and stored in Petri dishes (15 cm in diameter × 2 cm in height) lined with moistened filter paper. Larvae hatched after being transferred to glass tubes containing an artificial diet to sustain their development, with approximately 50 eggs placed in each tube. The insects were maintained in climate-controlled rooms at a temperature of 28 ± 1 °C, relative humidity of 70 ± 10, and a photoperiod of 12 h of light and 12 h of darkness.

### 2.2. Origin and Maintenance of Populations of the Parasitoid Cotesia flavipes

Two populations of the parasitoid *C. flavipes* were utilized in the experiments. One population was sourced through field collection from a sugarcane area in the county of Pradópolis, SP, Brazil (latitude: 21°21′23″, longitude: 48°03′48″) in April 2019. To obtain the parasitoids, pupae of *C. flavipes* collected from the field were brought to the Laboratory of Entomology of the University of Franca and monitored until adult emergence, resulting in 15 pupal masses. The parasitoids were collected from an area with no history of release to ensure the utilization of distinct strains of the parasitoid in the experiments. The other population of the parasitoid originated from the Laboratory of Entomology at Usina São Martinho. This population has been maintained in the laboratory at the plant since 1979 without the introduction of new individuals. Both populations were housed in the Laboratory of Entomology of the University of Franca for the purpose of conducting bioassays.

For the laboratory rearing and propagation of *C. flavipes*, 10 pupal masses from each population of parasitoids were individually placed in Petri dishes (9 cm in diameter × 2 cm in height). Upon emergence, females from each population were selected for parasitism. Fifth-instar larvae of *D. saccharalis* were parasitized, with only one female of *C. flavipes* allowed to parasitize each host [17]. Following parasitism, the *D. saccharalis* larvae were transferred to Petri dishes (9 cm in diameter × 2 cm in height) containing an artificial diet cut into pieces measuring 2 cm × 2 cm [21], facilitating the development of the parasitoid. Once *C. flavipes* pupae were formed, individuals were relocated to clean Petri dishes. The insects were maintained in climate-controlled rooms set at a temperature of 25 ± 1 °C, relative humidity of 70 ± 10, and a photoperiod of 12 h of light and 12 h of darkness. This process was repeated throughout the generations of the parasitoid for both studied populations.

### 2.3. Development and Offspring Production

The experiments were conducted at the Entomology Laboratory of the University of Franca, within a controlled environment room maintained at a temperature of 25 ± 1 °C, relative humidity of 70 ± 10%, and a photoperiod of 12 h of light and 12 h of darkness. Fifth-instar larvae of *D. saccharalis* were utilized for parasitization by female *C. flavipes* from both parasitoid populations. Each host was parasitized by a single female *C. flavipes*, following the previously described methodology [17]. The *D. saccharalis* larvae were parasitized once and then transferred to Petri dishes (8.5 cm in diameter; one larva per dish) containing a 2 cm^3^ portion of artificial diet to sustain the parasitized hosts. The insects were checked every 24 h to monitor the formation of parasitoid pupae. Once pupae were observed, they were transferred to Petri dishes (9 cm in diameter × 2 cm in height) within 24 h of formation. Pupal masses were monitored every 24 h to record adult emergence. The duration from egg to pupa (days) and the pupal period (days) were recorded. After the emergence and subsequent death of adults, the number of male and female offspring obtained from each parasitized host and the sex ratio were documented. Each treatment (population) was replicated 10 times, with each Petri dish serving as a replicate. Assessments of the biological characteristics of *C. flavipes* populations were conducted in the fifth, tenth, fifteenth, and twentieth generations of the parasitoid.

The sex ratio (SR) was calculated using the following formula:SR = (number of females)/(Number of females + number of males)(1)

### 2.4. Adult Survival

To assess the survival of adult parasitoids, 20 males and 20 females less than six hours old were carefully chosen, constituting 10 replicates for each treatment (*C. flavipes* populations). Each replication comprised two mating pairs housed in flat-bottomed test tubes measuring 2 cm in diameter × 8 cm in height. These tubes were sealed with PVC plastic film and provided no food, mimicking the conditions experienced in biofactories. Survival rates were monitored at six-hour intervals, starting from emergence until the death of all individuals. The experiments were conducted in a climate-controlled room maintained at a temperature of 25 ± 1 °C, under a photoperiod of 12 h of light/12 h of darkness, and with a relative humidity of 70 ± 10%.

Survival assessments of individuals from *C. flavipes* populations were conducted in the fifth, tenth, fifteenth, and twentieth generations of the parasitoid.

### 2.5. Flight Test

To assess the flight activity of *C. flavipes*, fifteen masses of pupae were utilized, with three masses placed in each flat-bottomed test tube (8.0 cm × 2.5 cm). These tubes were then positioned individually in chambers, resulting in approximately 150 adults per chamber. Each chamber consisted of a tubular container measuring 20 cm in height and 15 cm in diameter (PVC tube), with black cardboard lined with a strip of Biocontrole^®^ entomological glue (0.5 cm wide) attached to its inner surface, positioned six centimeters from the base. A transparent plastic sheet, slightly larger than the cylinder’s diameter (20 cm × 20 cm), was placed over the container. This plastic sheet was coated entirely with entomological glue on the side facing the chamber’s interior, following the methodology adapted from Trevisan [22] (Figure 1).

Once the adults emerged, they were allowed to move freely around the entire enclosure. Upon the observation of the mortality of all individuals, the test was concluded, and the adults were counted, categorizing them based on their location within the chamber. Insects that adhered to the internal glue band or were found at the bottom of the chamber were classified as walking insects, while those found adhered to the upper part of the chamber were classified as flying insects.

Five replicates were observed for each population of *C. flavipes*, with each chamber serving as a replicate. The insects were maintained in a climate-controlled room set at a temperature of 25 ± 1 °C, with a photoperiod of 12 h light and 12 h darkness, and a relative humidity of 70 ± 10%. The flight test was conducted using individuals from each parasitoid population in the tenth, fifteenth, and twentieth generations. However, it was not feasible to assess the flight activity of the parasitoids in the fifth generation due to the initial population size being insufficient, leading to the omission of these data.

### 2.6. Data Analyses

Data on the duration from egg to adult emergence, pupal period, number of offspring produced by *C. flavipes* from each parasitized host, and the sex ratio of parasitoids from each population were subjected to the Shapiro–Wilk test [23] to assess normality and the Bartlett test [24] to assess homogeneity of variance. When necessary, data were transformed (square root x + 0.5) to meet the assumptions of analysis of variance (ANOVA). Subsequently, ANOVA was conducted to identify potential differences between the main effects (populations and flight activity) and the interaction of factors (populations × flight activity), following a 2 (populations) × 2 (fliers and walkers) factorial design [25]. Post hoc comparisons of means were performed using the Tukey test (*p* < 0.05). All statistical analyses were conducted using SAS On Demand for Academics software version 3.82 (SAS Institute Inc., Cary, NC, USA).

The survival rate during the adult phase, initially with 20 females and 20 males per treatment, was determined by estimating the survival curve of parasitoids from various populations across different generations. The Kaplan–Meier method was applied using PROC LIFETEST in SAS software, with the Log-Rank test employed to compare survival rates between treatments.

The flight activity data (percentage of adults flying and walking) underwent normality and homogeneity of variance tests using the Shapiro–Wilk [23] and Bartlett [24] tests, respectively. The data were transformed (square root of x + 0.5) to fulfill the assumptions of analysis of variance (ANOVA). Subsequently, ANOVA was conducted to identify potential differences among the main effects (populations and flight activity) and their interaction (populations × flight activity) in a 2 (populations) × 2 (fliers and walkers) factorial design [25]. In the case of significance, the means were compared using the Chi-Square Test of Independence at a 5% significance level.

## 3. Results

### 3.1. Development and Offspring Production

The egg-to-pupa period differed significantly among the studied generations (F_1, 54_ = 180.75; *p* < 0.0001) and showed a significant interaction between generations and populations (F_1, 54_ = 14.75; *p* < 0.0001), while no significant difference was observed between populations (F_1, 54_ = 0.85; *p* = 0.3608). Specifically, in the laboratory population, this period was lengthier in generations 5 and 15 (14.0 days), whereas in the field population, it was consistently shorter across generations, notably in generation 20 (11.0 days) (Table 1).

Similarly, the pupal period significantly varied among generations (F_1, 54_ = 49.56; *p* < 0.0001), with no significant difference between populations (F_1, 54_ = 3.60; *p* = 0.0632), and no significant interaction between generations and populations (F_1, 54_ = 0.73; *p* = 0.5385). Notably, both laboratory and field populations exhibited the longest pupal period in generation 15 (Table 1).

A significant difference was observed in the number of offspring emerging from parasitized hosts between populations (F_1, 54_ = 8.73; *p* = 0.0003), generations (F_1, 54_ = 6.51; *p* = 0.0008), and in the interaction between generations and populations (F_1, 54_ = 4.44; *p* = 0.0074). Comparing the generations, the population maintained in the laboratory consistently exhibited similar values ranging from 32.0 to 55.9 parasitoids per host. In contrast, comparing the populations, the field population showed a lower number of adults emerging per host in the fifth and tenth generations. However, in the fifteenth and twentieth generations, it demonstrated a higher production of offspring per parasitized host (Table 2).

Significant differences were observed in the sex ratio between populations (F_1, 54_ = 29.97; *p* < 0.0001) and generations (F_1, 54_ = 8.89; *p* = 0.0002), but no significant difference was found in the interaction between generations and populations (F_1, 54_ = 0.91; *p* = 0.4425). When comparing generations, the laboratory population consistently exhibited similar values ranging from 0.3 to 0.7, indicating a relatively balanced sex ratio. In contrast, the field population showed higher proportions of females than males in the fifth, tenth, and twentieth generations. Moreover, comparing the populations, the laboratory population consistently demonstrated higher sex ratio values in the fifth, fifteenth, and twentieth generations (Table 2).

### 3.2. Adult Survival

Adult survival exceeded 80% in both *C. flavipes* populations up to approximately 25 h of age across all generations evaluated. Following this period, survival declined sharply, with all adults dying by 40 h in both the laboratory and field populations, consistently across generations (fifth generation: χ^2^ = 0.0706; *p* = 0.7905; tenth generation: χ^2^ = 1.7469; *p* = 0.1863; fifteenth generation: χ^2^ = 2.0684; *p* = 0.1504; twentieth generation: χ^2^ = 0.9877; *p* = 0.3203) (Figure 2).

### 3.3. Flight Test

The flight activity test results for *C. flavipes* populations indicated that both the laboratory-maintained population (χ^2^ = 12.39; *p* < 0.0001) and the field-derived population (χ^2^ = 27.42; *p* < 0.0001) exhibited significantly higher percentages of insects classified as walkers rather than fliers. In the laboratory population, the proportion of insects classified as fliers was 30.3%, 32.1%, and 32.3% in the tenth, fifteenth, and twentieth generations, respectively. Similarly, the field population showed 29.3%, 25.7%, and 30.0% of fliers in these same generations (Figure 3).

## 4. Discussion

This study aimed to assess the characteristics of a field-collected population of *C. flavipes* to incorporate individuals from this population into a laboratory-rearing system, enhancing overall quality. Biological traits, survival rates, and flight activity of the parasitoids were evaluated. The findings indicate that establishing a laboratory-rearing system for field-collected *C. flavipes* is feasible, though stabilization of the field-collected population requires several generations. The field-collected population stabilized in laboratory conditions from the tenth generation onward. Additionally, assessments of biological characteristics, longevity, and flight activity indicated comparable quality between the field-collected and laboratory populations once stabilization was achieved. Therefore, incorporating new individuals from the field can help increase the genetic diversity of laboratory populations without compromising quality.

Although we did not directly quantify the parasitization rate (i.e., the proportion of hosts effectively parasitized), the number of adults that successfully emerged from parasitized hosts was similar between the field-collected and laboratory-maintained populations. This finding suggests comparable parasitism performance and offspring viability in both populations. However, we recognize that a direct assessment of the parasitization rate would provide a more precise understanding of host acceptance and oviposition success. We, therefore, acknowledge this as a limitation of the current study and suggest it as an important aspect for future investigations.

The production of parasitoids inherently depends on the simultaneous cultivation of their hosts, requiring the management of two organisms to establish an effective natural enemy for a target pest [14]. In Brazil, the laboratory rearing of *C. flavipes* relies on the cultivation of its host, *D. saccharalis*, which is supported by a wide range of artificial diets [21]. However, this diversity in diets can significantly influence host quality, subsequently impacting the performance of the parasitoid [18,26]. Furthermore, contamination of the host may impair the flight ability of *C. flavipes* [27].

To ensure the success of biological control programs, it is critical to verify that essential operational procedures—such as laboratory asepsis and the employment of skilled labor—are consistently applied. Additionally, rigorous evaluation of host diets is necessary to guarantee the production of high-quality hosts and parasitoids capable of fulfilling their intended biological control roles. Specifically, the host must support the development of natural enemies that can effectively suppress pest populations.

These considerations are particularly relevant to our research, which examines a population of *C. flavipes* reared on hosts fed either artificial diets or sugarcane plants while the parasitoids remained in their natural field environment. Such investigations aim to elucidate the impact of host-rearing conditions on the performance and quality of parasitoids used in pest management.

Previous studies indicate that Brazilian populations of *C. flavipes* may exhibit variations in flight activity. Sipriano-Nascimento [28] assessed the flight activity of adult *C. flavipes* and found that some populations had over 50% of individuals classified as fliers, while others had less than 50% of flying individuals. In the present study, over 50% of individuals were classified as walkers. This observation may be attributed to inbreeding, as *C. flavipes* populations introduced since 1974 have been maintained in laboratory conditions without the addition of new individuals from the parasitoid’s place of origin.

Field studies are essential to evaluate the actual effectiveness of *C. flavipes* and potential “domestication” effects. Volpe et al. [19] reported a 26.5% reduction in dispersal capacity in *C. flavipes* compared to the originally introduced individuals [20]. Additionally, comparing the biology of parasitoids collected from different regions of Brazil has provided insight into the adaptations of specific strains to particular areas of the country [28]. Thus, it is recommended that quality control protocols be standardized and conducted periodically in laboratory facilities where *C. flavipes* is mass-reared, ensuring consistent records to monitor colony development. Prior research has demonstrated that the degenerative effects of inbreeding were not observed in the biology or survival of *C. flavipes* after 10 consecutive generations of strict inbreeding under laboratory conditions [17].

Future studies are needed to compare the Brazilian population of *C. flavipes* with populations from its native Indo-Australian region [29]. Such comparisons would help determine whether the Brazilian population has experienced quality loss after 50 years of laboratory rearing. However, this research faces significant challenges, primarily due to Brazilian regulatory requirements, which can be highly bureaucratic. For instance, introducing a parasitoid species into Brazil can take up to a year, complicating the logistics of this type of research. Nevertheless, addressing this question is critical for ensuring the ongoing success of the *C. flavipes* biological control program in Brazil, and thus, despite the challenges, such a study should be prioritized in the near future.

Biological control companies that mass-produce the parasitoid *C. flavipes* often face challenges maintaining consistently high-quality standards. Since *C. flavipes* is an exotic species originating near Pakistan and India, Brazilian lineages are prone to inbreeding over successive generations, which can lead to reduced aggressiveness and other undesirable traits. This study presents a new technique that enhances genetic diversity within parasitoid populations, thereby improving the quality of laboratory-reared parasitoids. The findings offer a practical approach that companies in the biological control industry can adopt to enhance the traits of mass-reared *C. flavipes*, optimizing their effectiveness in pest management.

*Cotesia flavipes*, like other species within the genus *Cotesia*, is known to be associated with a Polydnavirus (PDV), which plays a critical role in suppressing the immune response of the host and facilitating successful parasitism [30]. Although the current study focused on biological and behavioral parameters, we acknowledge that the long-term laboratory rearing of the *C. flavipes* population over several decades could have led to potential alterations in the symbiotic virus-host interaction, including changes in PDV structure, expression, or efficacy [31]. These factors were not evaluated here, and future studies incorporating genomic or transcriptomic analyses are needed to investigate whether PDV-associated mechanisms have evolved or remained stable under laboratory conditions [32]. Such investigations would provide valuable insights into the functional stability of key traits underlying parasitism success and could contribute to the optimization of mass-rearing programs for biological control.

These results indicate that field-collected *C. flavipes* populations may initially exhibit enhanced biological traits, such as shorter development time and increased offspring production per host, as observed in the twentieth generation (56.5 parasitoids/host). Although the present study did not evaluate hybrid progeny between laboratory and field populations, the observed differences support the hypothesis that introducing field-collected individuals into laboratory colonies may help maintain genetic diversity, improve colony vigor, and enhance the overall efficiency of mass-rearing programs. After ten generations under laboratory conditions, the field-collected population stabilized, and the quality of the individuals—based on biological traits, longevity, and flight activity—became comparable to that of the long-established laboratory population. Therefore, future studies should investigate the performance of hybrid progeny and evaluate whether periodic replenishment of laboratory colonies with field-collected individuals can provide long-term benefits for parasitoid production systems.

## Figures and Tables

**Figure 1 insects-16-00571-f001:**
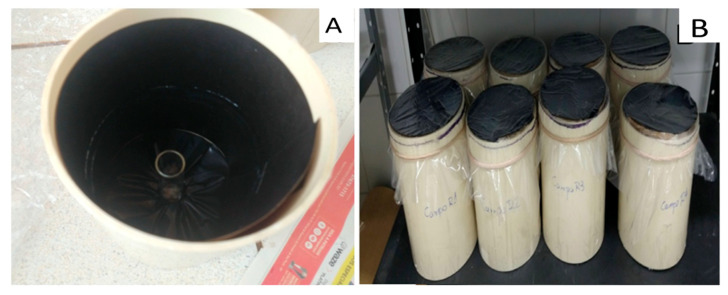
Chamber used for conducting a flight test of *Cotesia flavipes*. (**A**) Internal view of the chamber, and (**B**) external view.

**Figure 2 insects-16-00571-f002:**
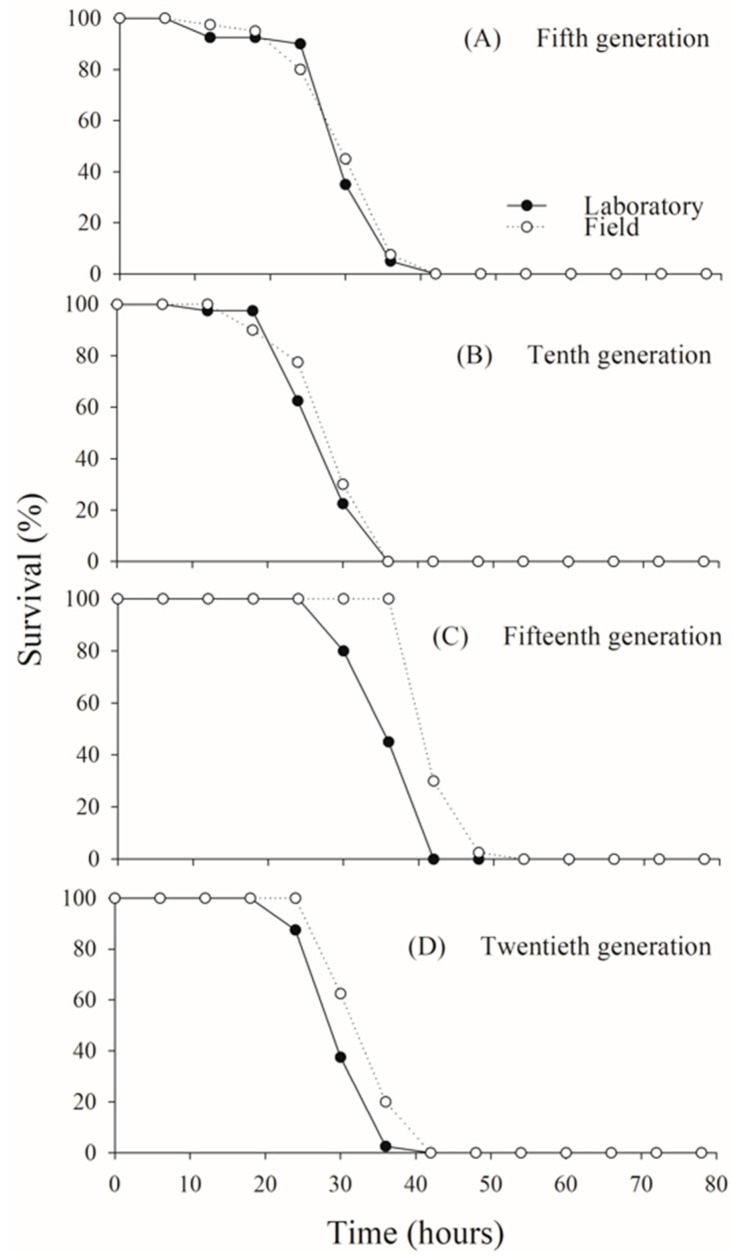
Adult survival rates in laboratory and field populations of *Cotesia flavipes* assessed under controlled laboratory conditions across multiple generations.

**Figure 3 insects-16-00571-f003:**
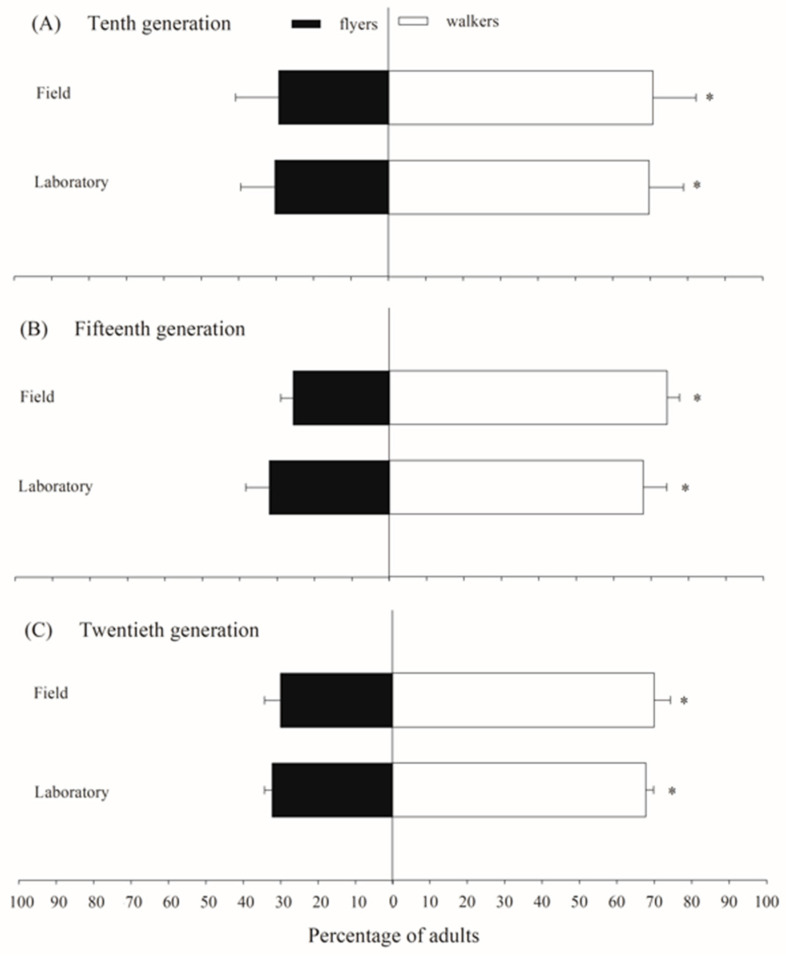
The proportion of adults classified as fliers and walkers in laboratory and field populations of *Cotesia flavipes* during flight activity tests across generations. The asterisk indicates a significant difference between flyers and walkers according to the Chi-Square test (*p* < 0.05).

**Table 1 insects-16-00571-t001:** Egg-to-pupa period and pupal period of *Cotesia flavipes* populations across generations.

Generations	Egg-to-Pupa (Days) ^a^		Pupal Period (Days) ^a^	
	Laboratory	Field	Laboratory	Field
05	14.0 ± 0.0 a	14.0 ± 0.0 a ^ns^	05.0 ± 0.0 c	05.0 ± 0.0 b ^ns^
10	11.4 ± 0.0 b	13.0 ± 0.0 b ^ns^	05.0 ± 0.3 c	05.0 ± 0.0 b ^ns^
15	14.0 ± 0.3 a	13.1 ± 0.1 b ^ns^	07.7 ± 0.3 a	07.3 ± 0.3 a ^ns^
20	10.8 ± 0.1 b	11.0 ± 0.0 c ^ns^	06.2 ± 0.3 b	05.6 ± 0.2 b ^ns^

^a^ Means ± standard error with the same letter in the column (generations) do not significantly differ according to the Tukey test (*p* > 0.05). “ns” denotes no significant difference between laboratory and field populations, while “*” indicates significance between populations.

**Table 2 insects-16-00571-t002:** Number of adults emerged per parasitized host, sex ratio and offspring of *Cotesia flavipes* populations across generations.

Generations	Adults/Host ^a^		Sex Ratio ^a^	
	Laboratory	Field	Laboratory	Field
05	55.9 ± 10.3 a	26.3 ± 04.5 b *	0.7 ± 0.0 a	0.5 ± 0.1 a *
10	36.0 ± 03.3 a	27.3 ± 03.6 b *	0.3 ± 0.0 a	0.2 ± 0.1 b ^ns^
15	50.6 ± 04.3 a	57.8 ± 07.4 a ^ns^	0.6 ± 0.0 a	0.3 ± 0.1 ab *
20	32.9 ± 03.4 a	56.5 ± 08.5 a *	0.6 ± 0.0 a	0.4 ± 0.0 ab *

^a^ Means ± standard error with the same letter in the column (generations) do not significantly differ according to the Tukey test (*p* > 0.05). “ns” denotes no significant difference between laboratory and field populations, while “*” indicates significance between populations.

## Data Availability

The raw data supporting the conclusions of this article will be made available by the authors on request.

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
