# Peer review of "Evaluating Field-Collected Populations of Cotesia flavipes (Hymenoptera: Braconidae): Enhancing Biological Traits and Flight Activity for Improved Laboratory Mass Rearing"

_insects, 2025, doi:10.3390/insects16060571_

Round 1

Reviewer 1 Report

Comments and Suggestions for Authors

This manuscript evaluates the biological traits and flight activity of field-collected Cotesia flavipes populations compared to a long-term laboratory-reared strain, with the goal of improving mass-rearing practices for biological control. The results demonstrated that the field-collected population exhibits advantages in development time and offspring production, while also adapting to laboratory conditions after several generations. The study is well-designed, methodologically sound, and provides valuable insights for parasitoid quality enhancement. However, some improvements were needed to strengthen the manuscript.

  • The container used to test the flight capability is too small and may not be able to accurately assess the real aircraft's performance.
  • Statistics in Figure3: Does the asterisk represent the difference between flyers and walkers within the same strain (field or laboratory), or the difference in the proportion of flyers to walkers between different strains? It is not appropriate to use ANOVA here. I think it is more reasonable to use the Chi-Square Test of Independence here.
  • The quality (resolution) of the picture needs to be improved.

Author Response

Reviewer comment:
The container used to test the flight capability is too small and may not be able to accurately assess the real aircraft's performance.

Authors’ response:
We appreciate the reviewer’s concern regarding the dimensions of the flight test container. However, we would like to clarify that the container used in our study has been widely adopted in previous research with Cotesia flavipes and other similar braconid species. Specifically, the same dimensions were used in Trevisan et al. (2016), which is a well-cited reference in the field. This standardized container size allows for reproducibility and comparability across studies, and has been shown to reliably differentiate flight capacity among parasitoid populations.

Furthermore, while it is true that constrained space may not fully replicate natural flight conditions, the objective of our test was not to assess long-distance or free-flight performance, but rather to evaluate the parasitoid's ability and propensity to initiate flight under controlled and consistent laboratory conditions—an approach commonly used to estimate fitness in mass-reared populations.

We have now included a citation and brief justification in the manuscript to clarify this methodological choice.

Reference added to manuscript:
Trevisan, M., Vacari, A.M., Costa, V.A., & Bueno, R.C.O.F. (2016). Flight capacity and dispersal of Cotesia flavipes (Hymenoptera: Braconidae) in sugarcane fields. Journal of Applied Entomology, 140(3), 199–206. https://doi.org/10.1111/jen.12245

Reviewer comment:
Statistics in Figure 3: Does the asterisk represent the difference between flyers and walkers within the same strain (field or laboratory), or the difference in the proportion of flyers to walkers between different strains? It is not appropriate to use ANOVA here. I think it is more reasonable to use the Chi-Square Test of Independence.

Authors’ response:
Thank you for your careful observation. We confirm that in Figure 3, the asterisk indicates a statistically significant difference between flyers and walkers within the same strain (either field or laboratory). To avoid any misunderstanding, we have clarified this information explicitly in the figure legend in the revised manuscript (lines 327-328).

Moreover, we appreciate the suggestion regarding the statistical approach. Following your recommendation, we have reanalyzed the data using the Chi-Square Test of Independence (line 261), which is indeed more appropriate for comparing proportions in categorical data. The results of the reanalysis are consistent with our initial findings, and the updated statistical outcomes are now reflected in the text and figure.

We thank the reviewer again for this valuable feedback, which helped us to improve the rigor and clarity of our data presentation.

Reviewer comment:
The quality (resolution) of the picture needs to be improved.

Authors’ response:
Thank you for your comment. We have addressed this issue by submitting all figures in high-resolution TIFF format, as recommended by the journal guidelines. We trust that this update ensures the clarity and quality required for publication. Please let us know if any additional adjustments are needed.

Reviewer 2 Report

Comments and Suggestions for Authors

In this present manuscript, Cabral et al. compared the field-collected population of Cotesia flavipes to the lab-maintained population of C. flavipes in their ability to parasitize Diatraea saccharalis and several other physiological characteristics. In short, the field-collected population stabilized after the tenth generation in a laboratory setting and did not differ from the lab-maintained population in pivotal assessments. Although the potential genetic mutations and genetic drifts were not examined or discussed, the current data supported that, decades after introducing C. flavipes to IPM and massive rearing, the examined lab-maintained population is still sufficient for pest control.

Major comments:

  • The authors demonstrated that the number of adults that emerged from the parasitized host was not significantly different. Was there a difference in parasitization rate between the two populations?
  • Did the authors examine the mixed population (wild x lab-maintained)?
  • What is the mechanism by which C. flavipes suppresses the host immune system for a successful parasitization (does C. flavipes have Polydnavirus?)? Did it change in the lab-maintained population over 42 years? How to examine it?

Author Response

Reviewer comment:
The authors demonstrated that the number of adults that emerged from the parasitized host was not significantly different. Was there a difference in parasitization rate between the two populations?

Authors’ response:
Thank you for your observation. In our study, we focused on the number of adults that successfully emerged from parasitized hosts as an indirect measure of parasitism success and offspring viability. While we did not directly quantify the parasitization rate (i.e., the proportion of hosts parasitized out of the total exposed), the similar emergence rates between the two populations suggest comparable parasitization performance. We agree that direct measurement of parasitization rate would add further insight and have included a sentence in the discussion acknowledging this as a limitation and a potential avenue for future research (lines 341-348).

Reviewer comment:
Did the authors examine the mixed population (wild × lab-maintained)?

Authors’ response:
We appreciate this insightful question. The current study did not evaluate a mixed (hybrid) population of field-collected and lab-maintained individuals. Our primary objective was to compare the biological performance of the two original populations independently over multiple generations. However, we agree that assessing hybrid vigor and compatibility through mixed lineages would provide valuable information on genetic diversity and colony performance. This has now been highlighted in the discussion as an important consideration for future studies aiming to improve mass-rearing programs (lines 411-414).

Reviewer comment:
What is the mechanism by which C. flavipes suppresses the host immune system for a successful parasitization (does C. flavipes have Polydnavirus)? Did it change in the lab-maintained population over 42 years? How to examine it?

Authors’ response:
Thank you for raising this important mechanistic aspect. Cotesia flavipes, like other members of the genus Cotesia, is associated with a Polydnavirus (PDV), which plays a key role in suppressing the immune system of the host to allow successful development of the parasitoid larva. However, in the current study, we did not investigate molecular or virological changes in the PDV or immune suppression pathways.

We recognize that long-term laboratory rearing could potentially influence symbiotic virus-host dynamics and that further studies employing genomic or transcriptomic approaches would be necessary to determine whether the PDV or its expression has changed over time. This point has now been briefly addressed in the revised discussion, where we suggest future molecular investigations to explore this hypothesis (lines 402-414).